# Association of Health Literacy and Nutritional Status Assessment with Glycemic Control in Adults with Type 2 Diabetes Mellitus

**DOI:** 10.3390/nu12103152

**Published:** 2020-10-15

**Authors:** Saman Agad Hashim, Mohd Yusof Barakatun-Nisak, Hazizi Abu Saad, Suriani Ismail, Osama Hamdy, Abbas Ali Mansour

**Affiliations:** 1Department of Dietetics, Faculty of Medicine and Health Sciences, Universiti Putra Malaysia, Serdang 43400, Malaysia; samman_79@yahoo.com; 2Sader Teaching Hospital, Basrah Health Directorate, Ministry of Health, Basrah 61001, Iraq; 3Institute for Social Sciences Studies, Universiti Putra Malaysia, Serdang 43400, Malaysia; 4Department of Nutrition, Faculty of Medicine and Health Sciences, Universiti Putra Malaysia, Serdang 43400, Malaysia; hazizi@upm.edu.my; 5Department of Community Health, Faculty of Medicine and Health Sciences, Universiti Putra Malaysia, Serdang 43400, Malaysia; si_suriani@upm.edu.my; 6Joslin Diabetes Centre, Harvard Medical School, Boston, MA 02215, USA; osama.hamdy@joslin.harvard.edu; 7Faiha Specialized Diabetes, Endocrine, and Metabolism Center (FDEMC), University of Basrah, Basrah 61001, Iraq; abbas.mansour@fdemc.iq

**Keywords:** health literacy, S-TOFHLA, Type 2 Diabetes Mellitus, glycemic control, Iraq

## Abstract

While the role of medical and nutrition factors on glycemic control among adults with type 2 diabetes mellitus (T2DM) has been well-established, the association between health literacy (H.L.) and glycemic control is inconsistent. This study aims to determine the association of H.L. and nutritional status assessments with glycemic control in adults with type 2 diabetes mellitus. A total of 280 T2DM respondents (mean (SD) age = 49.7 (10.3) years, Glycated hemoglobin (HbA1c) = 9.9 (2.6) %, and Body Mass Index = 32.7 (15.1) kg/m^2^) were included in this study. A short-form Test of Functional Health Literacy in Adults (S-TOFHLA) assessed the H.L. levels. Nutritional status assessments included client history, glycemic control, anthropometric, and biochemical data. The mean (S.D.) H.L. score was 45.7 (24.6), with 56% of the respondents had inadequate H.L. Inadequate H.L. was more common among those females; housewives, low education, received oral antidiabetic therapy, and shorter diabetes duration. Respondents with inadequate H.L. were significantly older and had higher HbA1c than those with marginal and adequate H.L. Meanwhile, respondents with inadequate and marginal H.L. levels had significantly higher total cholesterol, LDL-cholesterol, and systolic blood pressure than the respondents with adequate H.L. Low H.L. scores, self-employment status, received dual antidiabetic therapy (insulin with oral agents), received insulin alone, and had higher fasting blood glucose explained about 21% of the total variation in HbA1c (adjusted R^2^ = 0.21; *p* < 0.001). Respondents with inadequate H.L. had poor glycemic control. The H.L. scores, together with nutritional status assessments, were the factors that predicted poor glycemic control among adults with T2DM.

## 1. Introduction

Type 2 diabetes mellitus (T2DM) is a global public health concern, and the Middle East region is not spared with the second top diabetes rate in the world. About 10% of the patients with T2DM were in the Middle East region, with almost 50% of them were undiagnosed [1]. In Iraq, the prevalence of T2DM was about 15%, and the highest was recorded in Basrah City (19.7%) [2]. Among those with T2DM, poor glycemic control was highly prevalent, with about 86.2% of T2DM had an HbA1c of more than 7% [3], which is consistent with the other studies worldwide [4,5]. About 40–60% of patients worldwide still have poorly controlled diabetes [6,7], suggesting the critical needs to understand the factors influencing glycemic control.

In the literature, studies reported various factors contributing to poor glycemic control among T2DM patients, including diabetes duration, diabetes treatment, medical background, and nutritional status [2,8,9]. Although different studies have been conducted in this area, the results are inconsistent and vary across countries as well as between other ethnic groups [6,10]. Furthermore, the non-clinical factors, in particular, the concept of health literacy (H.L.), has emerged as an influential factor in the health outcomes for patients with complex chronic diseases, including T2DM [11].

H.L. is principally defined as ‘the degree to which individuals can obtain, process, understand and communicate about health-related information needed to make informed health decisions’ [12]. There are various instruments used to assess H.L. in the diabetes population, such as the Rapid Estimate of Adult Literacy (REALM) [13], Diabetes Numeracy Test (D.N.T.) [14], and Test of Functional Health Literacy in Adults (S-TOFHLA) [15], of which S-TOFHLA was the most commonly used and considered a suitable tool to evaluate H.L. among adults with T2DM [16]. S-TOFHLA is self-administered to assess reading comprehension by determining a patient’s ability to read phrases and passages containing numbers using real objects in health care settings [17]. On the other hand, the REALM recognizes patients with low reading levels without assessing the words’ understanding [13], and D.N.T. focusing only on the aspects of numeracy skills [14]. A low H.L. level is typically reported worldwide, ranging from 7.3 to 82%, lowest in Switzerland, and the highest in Taiwan [18]. Nonetheless, little research exists to understand the H.L. levels in patients with T2DM in Iraq.

A study by Abdul-Hasan & Yassin 2018 identified that Iraqi T2DM patients had inadequate H.L. levels with the mean scores of 43.3. In this study, H.L. scores were significantly associated with poor glycemic control and diabetes complications [19]. However, the mentioned study used different instruments that the author self-developed, which was not adequately validated. Also, the tool has not covered the whole spectrum of H.L. measures. S-TOFHLA is considered the best at present as it was validated in the Arabic language and measured numeracy and reading comprehension simultaneously [15,20]. A study conducted in Iraq used the S-TOFHLA to assess H.L. among customers who attended the pharmacies in two different cities. The study identified about 17% had low levels of H.L., which was lower as compared to the studies in Western countries [21]. The data indicate the critical needs of assessing H.L. among patients with T2DM and its association with glycemic control. Data on H.L. among T2DM is vital in designing the appropriate intervention tailored to Iraqi due to the rising number of diabetes in Iraq. Adequate H.L. promotes active participation in self-care activities leading to better glycemic control than those without adequate H.L. levels [22]. The role of medical and nutrition factors on glycemic control among adults with type 2 diabetes mellitus (T2DM) has been well-established. Nonetheless, the association between H.L. and glycemic control is still inconsistent. Therefore, this study aims to determine the association of H.L. and nutritional status assessments with glycemic control in adults with T2DM.

## 2. Methods

### 2.1. Study Design and Respondents

This cross-sectional study was conducted between January and April 2019 among respondents with T2DM at Faiha Specialized Diabetes, Endocrine, and Metabolism Centre (FDEMC), Basrah, Iraq. Respondents aged 20-64 years with a confirmed diagnosis of T2DM and had no hearing or vision impairment were included in the study. This study excluded those patients with type 1 diabetes, gestational diabetes or those with severe illness such as cancer and kidney failure. The institutional ethics committee approved the study (UPM/TNCPI/RMC/1.4.18.2 (JKEUPM)), and (FDEMC/ 56/35/22)), and all respondents provided their written consent before study enrolment.

### 2.2. Sampling and Sample Size

The respondents were sampled using a random sampling technique from the hospital list. Daily, about 100 patients would be attending Faiha Specialized Diabetes, Endocrine, and Metabolism Centre. According to the sample size, eight respondents were needed to be recruited daily during routine visits. Hence, we included every 12th patient from the list as a starting point generated by SPSS. The sample size was determined using a specific formula used in a cross-sectional study conducted by Daniel [23], based on the prevalence of T2DM (19.7%) reported among respondents with T2DM in Basrah City [2] as the following:
*n* = (Z1 − a/2) 2 *p* (1 − *p*)/d^2^

With a 95% confidence level and the adjustment of the 20% drop-out [23], a minimum of 280 respondents would be recruited in the study.

### 2.3. Data Collection and Measurements

Data were collected using a standardized questionnaire for nutritional status assessments that included client history, glycemic control, anthropometry, and biochemical data, as well as H.L. measurements. The client history and H.L. were collected from all the respondents when they attended Faiha Specialized Diabetes, Endocrine, and Metabolism Centre for their routine visits. The data were collected using a one-to-one interview in a particular room for about 10–15 min. The latest anthropometry and biochemical data were obtained from the medical records.

#### 2.3.1. Client History

The client’s history included age, sex, marital status, educational level, employment status, and monthly household income were obtained using a standardized questionnaire. Data about the duration of T2DM, treatment modalities, family history, and the presence of comorbid conditions were obtained from medical records.

#### 2.3.2. Health Literacy Scale

H.L. was assessed using the Iraqi (Arabic) version of the S-TOFHLA with reported reliability of 0.65 for numeracy and 0.89 for reading scores. The questionnaire was validated in Iraqi populations [21]. S-TOFHLA consisted of both reading comprehension and numeracy competency sections with a total score of 100 points. The reading comprehension test consisted of two passages, with a total of 35 items. Each passage has few words deleted. Respondents must choose the best words to complete the sentences for each blank space using four given options. The total score for reading comprehension was 70 points. The numeracy tests included four questions that assessed the literacy-related to medications, levels of blood glucose, and appointment dates [24]. The total score for numeracy was 30 points. The score between 0 and 53 indicated as inadequate H.L.; between 54 and 66 as a marginal H.L., and those scored between 67 and 100 are recognized as having adequate H.L. [24].

#### 2.3.3. Anthropometry and Biochemical Data

The body weight and height of the respondents were assessed using a standardized methodology. The weight was measured in a standing position using a digital calibrated weighing scale (SECA; British Indicators, London U.K.) to the nearest 0.1 kg in light clothing without shoes, watches, wallets, jewels, and other accessories that would affect the accuracy of the measurement. The height was measured without shoes or slippers using a height scale (SECA; British Indicators Ltd., London, UK) to the nearest 0.1 cm. The respondents asked to stand straight on the stadiometer’s floorboard while their backs were placed against the stadiometer’s vertical backboard. Body weight and height were used to calculate the body mass index (B.M.I.), and the bodyweight status was classified using the World Health Organization standard. [25]. B.M.I. was classified as normal if B.M.I. was 18.5–24.9 kg/m^2^, while overweight and obesity if B.M.I.s were 25–29.9 kg/m^2^, and ≥30 kg/m^2^, respectively [25].

Blood pressure and recent blood results for hemoglobin A1C (HbA1c), fasting blood glucose (F.B.G.), total cholesterol (T.C.), low-density lipoprotein cholesterol (LDL-C), high-density lipoprotein cholesterol (HDL-C), and triglycerides (T.G.) were also obtained from medical records with the optimal ranges were based on the American Diabetes Associations [26].

### 2.4. Statistical Analyses

The SPSS software (I.B.M. Corp., Armonk, NY, USA, 2019) was used for data analyses, and the level of significance was set at *p* < 0.05. Descriptive data were presented using means (S.D.) for continuous variables while categorical data shown using percentages and frequencies. Data were checked for normality using the Kolmogorov Smirnov statistics. HbA1c scores transformed to be normally distributed using a two-step approach [27]. Respondents were grouped according to their H.L. levels, and the mean differences were compared using one-way analyses of variance (ANOVA) for continuous data Chi-square or Fisher’s exact test for categorical data. Tukey’s post hoc test was conducted to explore the differences between multiple groups. Pearson correlation was used to assess the correlation between continuous, dichotomous variables, and HbA1c. Categorical variables were converted to dichotomous (0, 1) and included in Pearson correlation coefficients.

The association of H.L. and nutritional status assessments with glycemic control as analyzed using multiple linear regressions with a stepwise method. We imputed the variables into the models when *p* < 0.20 in the bivariate analysis [15]. In multiple linear regression analyses, the categorical variables with two categories were coded using the dummy coding system, such as sex (0 = male, 1 = female). For more than two levels, such as treatment modalities, we coded into four categories in the following order; where 0 = oral antidiabetic therapy (reference category), 1 = insulin therapy only, 2 = dual antidiabetic therapy (insulin and oral antidiabetic therapy), and 3 = diet alone. The categorization is necessary to maintain and include all categories of the variable in the predicting models. A 95% confidence interval was established. The variance inflation factor (V.I.F.) was 1.07 indicated that the predictors are weakly correlated but not enough to be overly concerned about it. Therefore, multicollinearity is not a problem in the current study.

## 3. Results

### 3.1. Recruitment of Respondents

A total of 665 patients from the database were screened, of which 329 met the study criteria. The main reasons for being excluded from the study included type 1 diabetes, gestational diabetes, hearing or vision problems, renal failure, or having severe illnesses like cancers. We invited all of them, but 329 respondents responded by, of which 310 of them agreed to participate and signed the consent form. The study included a total of 280 respondents who provided 85% of the response rates (Figure 1).

### 3.2. Client History and Health Literacy Levels

Respondents in this study were in their 50s’ with more than half were females (55%), and nearly half (41.8%) had low education and diagnosed with diabetes for 1–5 years (46.4%) (Table 1). The mean (SD) HL score was 45.7 (24.6), with nearly 80% (77.4%) had either inadequate (55.6%) or marginal HL (20.8%) levels. Only 23.5% of the respondents had adequate H.L. The average mean (S.D.) H.L. scores for each category were 27.62 (15.92) for inadequate H.L., 60.53 (15.92) for marginal H.L., and 75.89 (9.46) for adequate H.L. Respondents with inadequate H.L. had significantly lower S-TOFHLA scores for numeracy and reading skills. They spent more time taking the test. Inadequate H.L. levels were more common among older and female respondents; housewives, they attained lower than middle school, had a shorter diabetes duration, and were mostly on oral agents. Respondents with inadequate H.L. levels were significantly older than those with marginal and adequate H.L. levels (*p* < 0.05).

### 3.3. Anthropometry, Biochemical, and Health Literacy Levels

The average HbA1c was 10.0 (2.7%), with 86.8% had poor glycemic control indicated as HbA1c of >7%. Respondents with inadequate HL had a significantly higher HbA1c levels (Mean (SD) = 10.6 (2.6%)) than those with marginal and adequate HL (*p* < 0.001). On the other hand, respondents with adequate HL had significantly better total cholesterol levels (mean (SD) = 9.4 (2.6 mmol/L)), LDL-cholesterol (mean (SD) = 6.4 (2.1 mmol/L)), and systolic blood pressure (mean (SD) = 135.3 (25.8 mmHg)) than those respondents who had inadequate and marginal HL. Mean Body Mass Index of respondents was within obesity (mean (SD) = 32.66 (15.11) kg/m^2^) (Table 2).

### 3.4. Association of Health Literacy and Nutritional Status Assessments with Glycemic Control

Table 3 shows a total of 20 variables were associated with HbA1c levels at the bivariate levels. These included age, income, presence with co-morbidities, B.M.I., F.B.G., T.C., diastolic B.P., H.L. score, oral antidiabetic therapy, dual antidiabetic therapy (insulin with oral agents), insulin therapy only, diet alone, low education, middle school, high school, and above, employed, retired, self-employed, 1–5 years present with T2DM, and 6–10 years present with T2DM.

### 3.5. Factors Predicting Glycemic Control (HbA1c)

Table 4 shows the factors that significantly predicted the HbA1c levels included HL score, self-employment, dual anti-diabetic therapy, insulin alone, and FBG (Adjusted R^2^ = 0.21, F (5, 279) = 15.698, *p* < 0.001). These variables explained 21% of the variability of HbA1c in this study. Among other factors in the model, the H.L. score contributed strongly to the HbA1c levels. For each 1-point decreased in the HL score, the HbA1c value increased by 0.32 (*p* < 0.001).

## 4. Discussion

This study shows that the majority of adults with T2DM had inadequate H.L. levels (56%). This study confirmed that inadequate H.L. contributed to poor glycemic control, together with nutritional status assessments, including self-employment, dual antidiabetic therapy, insulin alone, and F.B.G. These factors explained 21% of the variability in HbA1c level. H.L. is an essential factor that predicted poor HbA1c. In diabetes, patients are required to understand the printed information, oral communication, and numeracy to practice self-diabetes care. All of these skills required adequate H.L. to ensure they can quickly navigate the healthcare system that facilitates to achieve optimal glycemic control [28,29].

The current result was similar to other studies conducted among respondents with T2DM in Brazil [30], Pakistan [31], and Iran [32]. All of these studies used S-TOFHLA to assess H.L. S-TOFHLA was the most commonly used and considered a suitable tool to evaluate H.L. among adults with T2DM [16]. S-TOFHLA can be self-administered to assess reading comprehension by determining a patient’s ability to read phrases and passages containing numbers using real objects in health care settings [17]. However, the finding was inconsistent with a study conducted in Saudi Arabia among respondents with T2DM. The study used S-TOFHLA, the same tool as in the current study to assess H.L. and found that H.L. was not associated with poor glycemic control but increased B.M.I. [22]. The discrepancy could be due to the difference in the health system in Saudi Arabia compared to Iraq, which explains more patients who had adequate H.L. and glycemic control [22] than the current study.

Self-employed respondents had a higher HbA1c than those in the employee category. The results were not consistent with previous studies [18,33]. The employment status did not predict glycemic control among Moroccans patients with T2DM [4]. Nevertheless, a study conducted among 325 adults with T2DM attending in Jimma University Teaching Hospital in Ethiopia showed that farmers with diabetes have inadequate glycemic control compared with unemployed patients. Self-employment was associated with more extensive social networks but had more stresses [34].

Besides, dual antidiabetic therapy and insulin therapy alone were significantly associated with poor glycemic control in the current study compared to those on the oral antidiabetic drug category. This finding is in line with other studies conducted among respondents with T2DM in Malaysia [35,36], Singapore [37], and Jordan [38]. A study conducted in Basrah city, Iraq, reported that insulin treatment was a significant factor of an increased risk of poor glycemic control among respondents with T2DM [3]. Lifestyle modification is the first-line therapy in diabetes management. If the optimal glycemic control is unable to achieve, single or dual oral antidiabetic therapy can be given. As diabetes progresses and the target glycemic control is even challenging to achieve, insulin injection would be administered [26]. Thus, that is consistent with our results to show that those with more complex diabetes treatments are the one who had poor glycemic control.

A higher F.B.G. was associated with poor glycemic control. The finding coincides with other studies conducted in Iran among 604 patients with T2DM. They found a strong association between HbA1c and F.B.G. [39]. The similarity in findings could be explained by the lack of control of F.B.G. in daily life lead to poor glycemic control among patients with T2DM.

Respondents with inadequate H.L. were housewives, attained lower education levels, shorter diabetes duration, and received oral antidiabetic therapy alone. The scenario has been commonly observed in previous studies conducted among respondents with T2DM as most of the housewives have no or less education, which difficult for them to be employed [31]. Low education is related to inadequate H.L. among individuals with T2DM because the specific skills are required to understand the printed information, oral communication, and numeracy to practice self-diabetes care. The ability to properly self-manage diabetes would facilitate optimal glycemic control [28,29].

This study observed that respondents with adequate H.L. achieved optimal glycemic control and cardiovascular risk controls because they can understand necessary health information to make informed health decisions [40]. This finding shares several similarities with previous results in Spain [41], the United States [15], Brazil [42], and Saudi Arabia [22]. The studies mentioned above found a significant association between adequate H.L. and good HbA1c or diabetes-related outcomes. Respondents with an adequate H.L. usually participate in continuous self-care activities, contributing to consistencies in lifestyle and treatment outcomes, as evidenced by several studies [15,20,22]. Therefore, adequate H.L. makes it easy to transfer important health information to diabetes respondents to achieve better glycemic control and avoid diabetes complications [43]. Other nutritional status parameters, including dietary intake and physical activity level, may enhance the prediction to HbA1c. Still, these factors did not include in this current study, which warrants future investigation on this aspect.

### Strength and Limitations

This study used simple random sampling to avoid selection bias. Besides, this study was the first study conducted in Iraq to assess the association of H.L. with glycemic control among respondents with T2DM, using the S-TOFHLA questionnaire. However, little experience with the S-TOFHLA scale among respondents with T2DM poses an obstacle to the authors. Then, illiterate respondents cannot participate in this study. Lastly, the study was conducted at a single hospital where all respondents were living in one city.

## 5. Conclusions

The majority of respondents with T2DM in Basrah, Iraq, had poor glycemic control and inadequate H.L. Respondents with inadequate H.L. levels were significantly older and had a higher HbA1c than those with marginal and adequate H.L. levels. Meanwhile, respondents with inadequate and marginal H.L. levels had significantly higher total cholesterol, LDL-cholesterol, and systolic blood pressure than the respondents with adequate H.L. levels. Low H.L. scores and nutritional status assessments, including self-employment, dual antidiabetic therapy, insulin alone, and elevated F.B.G. were predictors of poor glycemic control among Iraqi respondents with T2DM. Educational interventions are required to improve H.L. that translates into better glycemic control among diabetes.

## Figures and Tables

**Figure 1 nutrients-12-03152-f001:**
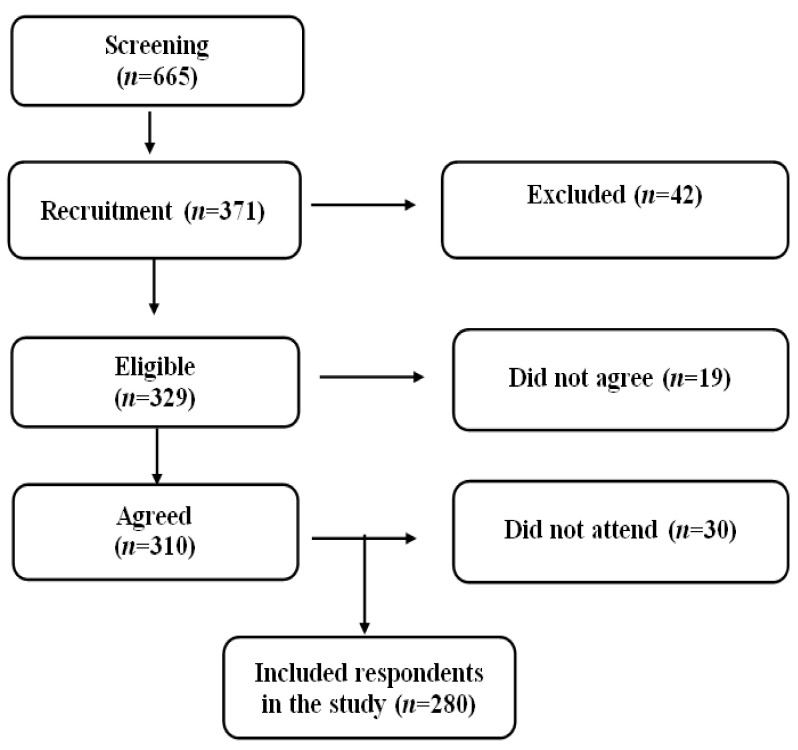
Screening and Recruitment of the Study Respondents.

**Table 1 nutrients-12-03152-t001:** Socio-demographic and medical status of respondents according to the health literacy levels.

Variables	Health Literacy Levels			Total (*n* = 280) Mean (SD)	*p*- Value ^†^
	Inadequate (*n* = 156, 56.7%) Mean (SD)	Marginal (*n* = 58; 20.7%) Mean (SD)	Adequate (*n* = 66; 23.5%) Mean (SD)		
Age (years)	50.02 (10.54) ^a^	49.49 (7.16) ^a^	46.37 (11.34) ^b^	49.65 (10.29)	0.05
Monthly income (USD)	825.16	892.96	943.01	867.07	0.755
	(1306.69)	(621.12)	(919.07)	(1107.49)	
S-TOFHLA numeracy ^ê^	8.55 (7.04) ^a^	16.29 (4.89) ^b^	25.00 (6.22) ^c^	14.03 (9.37)	<0.001
S-TOFHLA reading ^ĝ^	19.31 (11.98) ^a^	44.52 (6.10) ^b^	50.80 (8.55) ^c^	31.96 (17.62)	<0.001
S-TOFHLA total	27.62 (15.92) ^a^	60.53(15.92) ^b^	75.89 (9.46) ^c^	45.82 (24.68)	<0.001
S-TOFHLA time (min)	22.50 (2.13) ^a^	20.93 (2.67) ^b^	17.38 (2.80)^c^	20.97 (3.19)	<0.001
**Variables**	***n*** **(%)**	***n*** **(%)**	***n*** **(%)**	***n*** **(%)**	***p*** **-Value ***
Sex					<0.001
Male	57 (36.5)	27 (46.6)	43 (65.2)	127 (45.4)	
Female	99 (63.5)	31 (53.4)	23 (34.8)	153 (54.6)	
Marital status					0.757 ^Ƒ^
Single	18 (11.5)	6 (10.3)	6 (9.1)	30 (10.7)	
Married	135 (86.5)	49 (84.5)	58 (87.9)	242 (86.4)	
Widowed (widower)	3 (1.9)	3 (5.2)	2 (3.0)	8 (2.9)	
Education level					<0.001 *
Low	84 (53.8)	23 (39.7)	10 (15.2)	117 (41.8)	
Middle school	37 (23.7)	16 (27.6)	28 (42.4)	81(28.9)	
High school and above	35 (22.4)	19 (32.8)	28 (42.4)	82 (29.3)	
Employment					<0.001 ^Ƒ^
Employee	30 (19.2)	23 (39.7)	34 (51.5)	87 (31.1)	
Retired	17 (10.9)	2 (3.4)	6 (9.1)	25 (8.9)	
Self-employed	22 (14.1)	4 (6.9)	11 (16.7)	37 (13.2)	
Housewife	82 (52.6)	25 (43.1)	11 (16.7)	118 (42.1)	
Unemployed	5 (3.2)	4 (6.9)	4 (6.1)	13 (4.6)	
Family History of D.M.					0.268 *
Yes	130 (83.3)	43 (74.1)	51 (77.3)	224 (80.0)	
No	26 (16.7)	15 (25.9)	15 (22.7)	56 (20.0)	
Years with Diabetes					0.005 *
1–5 years	71 (45.5)	24 (41.4)	35 (53.0)	130 (46.4)	
6–10 years	65 (41.7)	17 (29.3)	14 (21.2)	96 (34.3)	
11 years and more	20 (12.8)	17 (29.3)	17 (25.8)	54 (19.3)	
Treatment Modalities					< 0.001 ^Ƒ^
Oral antidiabetic therapy	73 (46.8)	31 (53.4)	19 (28.8)	123 (43.9)	
Insulin therapy only	10 (6.4)	8 (13.8)	19 (28.8)	37 (13.2)	
Dual antidiabetic therapy	68 (43.6)	19 (32.8)	24 (36.4)	111 (39.6)	
Diet alone	5 (3.2)	0 (0.0)	4 (6.1)	9 (3.2)	
Co-morbidities					0.339 *
Yes	103 (66.0)	44 (75.9)	43 (65.2)	190 (67.9)	
No	53 (34.0)	14 (24.1)	23 (34.8)	90 (32.1)	

* The Chi-square test or **^Ƒ^** Fisher Exact test was used for categorical variables; ^†^: scored out of 30; ^ĝ^: scored out of 70; ª one-way analysis of variance (ANOVA), for means of continuous variables; N (%). Data are presented as number and percentage, S.D. = standard deviation, US$ = United States Dollar, D.M. = diabetes mellitus. ^a,b,c^ = means in a row with different superscript letters are significantly different among groups with post hoc tests. S-TOFHLA = short-form Test of Functional Health Literacy in Adults. The *p*-value is significant at the 0.05 level.

**Table 2 nutrients-12-03152-t002:** Anthropometrics and biochemical data of respondents according to the health literacy levels.

Characteristics	Health Literacy Level			Total (*n* = 280, 100%) Mean (SD)	*p*-Value
	Inadequate (*n* = 156, 56.7%) Mean (SD)	Marginal (*n* = 58; 20.7%) Mean (SD)	Adequate (*n* = 66; 23.5%) Mean (SD)		
Body Mass Index (kg/m^2^)	32.99 (13.22)	31.11 (13.77)	33.33 (19.9)	32.66 (15.11)	0.680
HbA1c (%)	10.63 (2.61) ^a^	9.02 (2.03) ^b^	9.13 (2.34) ^b^	9.99 (2.66)	<0.001
FBG (mg/dL)	269.20 (116.14)	246.13 (106.64)	235.35 (111.33)	256.44 (113.68)	0.09
Total Cholesterol (mg/dL)	196.75 (42.24) ^a^	199.03 (54.23) ^a^	168.34 (46.43) ^b^	190.55 (42.2)	<0.001
LDL-C (mg/dL)	143.72 (47.41) ^a^	139.95 (53.01) ^a^	114.42 (37.22) ^b^	136.00 (48.00)	<0.001
HDL-C (mg/dL)	44.33 (20.53)	44.42 (9.46)	42.54 (12.03)	43.99 (16.99)	0.760
Triglyceride (mg/dL)	214.43 (110.27)	215.84 (102.52)	190.03 (105.33)	208.99 (107.66)	0.260
Systolic BP (mmHg)	138.23 (20.63) ^a^	138.78 (24.78) ^a^	125.15 (33.87) ^b^	135.33 (25.77)	0.001
Diastolic BP (mmHg)	83.24 (10.8)	88.06 (20.87)	83.8 (10.48)	84.333 (13.55)	0.060

Note: one-way ANOVA was conducted; n (%). Data are presented as number and percentage. SD = standard deviation; HbA1c = Glycated hemoglobin; FBG = fasting blood glucose; mg/dL = milligram/deciliter; LDL-C = low-density lipoprotein cholesterol; HDL-C = high-density lipoprotein cholesterol; BP = blood pressure; ^a,b^ = means in a row with different superscript letters are significantly different among groups with post hoc test. The *p*-value is significant at the 0.05 level.

**Table 3 nutrients-12-03152-t003:** Factors associated with glycemic control (HbA1c) of the respondents.

No.	Variables	*r*	*p*-Value
1	Age (years)	−0.094	0.11
2	Income (USD)	−0.272	<0.001
3	Presence of the co-morbidities	0.118	0.05
4	Body mass index (kg/m^2^)	−0.118	0.05
5	Fasting blood glucose (mg/dL)	0.153	0.01
6	Total cholesterol (mg/dL)	0.158	0.008
7	Diastolic blood pressure (mmHg)	−0.107	0.07
8	Health literacy score	−0.301	<0.001
9	Oral antidiabetic drugs therapy	−0.288	<0.001
10	Dual antidiabetic therapy	0.241	<0.001
11	Insulin therapy only	0.081	0.001
12	Diet alone	0.081	0.18
13	Low education	0.190	0.001
14	Middle school	−0.121	0.04
15	High school & above	−0.085	0.15
16	Employee	−0.186	0.002
17	Retired	−0.091	0.13
18	Self-employed	0.242	0.001
19	1–5 years present with diabetes	−0.158	0.008
20	6–10 years present with diabetes	0.110	0.06

r = Pearson correlation coefficients; HbA1c = Glycated hemoglobin. The *p*-value is significant at the 0.05 level.

**Table 4 nutrients-12-03152-t004:** Predictive factors for HbA1c levels.

Variables	B (95% CI)	β	R^2^	Adjusted R^2^	t	*p*-Value
HL score using S-TOFHLA	−0.02	−0.32	0.09	0.09	−5.90	<0.001
Employment vs. employees						
Self-employed	0.87	0.19	0.14	0.14	3.44	0.001
Treatment modalities vs. oral antidiabetic therapy						
Dual anti-diabetic therapy	0.82	0.25	0.18	0.17	4.52	<0.001
Insulin therapy only	0.89	0.19	0.21	0.20	3.20	0.001
Fasting blood glucose	0.001	0.11	0.22	0.21	1.97	0.05

S-TOFHLA = A short-form Test of Functional Health Literacy in Adults. B = unstandardized Coefficients, CI = confidence interval, β = standardized Coefficients, R^2^ = the proportion of variance in the criterion, t = t-statistic, HL = health literacy. The *p*-value is significant at the 0.05 level.

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
