# Peer review of "Association of Health Literacy and Nutritional Status Assessment with Glycemic Control in Adults with Type 2 Diabetes Mellitus"

_nutrients, 2020, doi:10.3390/nu12103152_

Round 1

Reviewer 1 Report

The article contains a lot of observations but is not a full study and the discussion/conclusions are overemphasizing the correlation of low health literacy and not controlled diabetes quite ignoring other components or trying to explain a possible correlation.

I have some suggestion for the authors:

Line 3: the title
please, change “Other Diabetes-related Factors” with something more precise

Lines 83-84

“Respondents aged between 20 and 64 years old and without hearing or vision impairment and severe illnesses like cancers and renal failure were included in the study. They were excluded if they had type 1 diabetes, gestational diabetes, renal failure, or underwent active treatment for cancers.”
these two phrases should be rephrased in one clearer, simple and not redounded sentence.

Lines 91-93:
so you have chosen the twelfth subject counted from a random number for a total of eight individuals (samples) per day? please rephrase your sentences

Line 93-94: “The sample size was determined using a specific formula (ref###) for a cross-sectional survey...”
please cite and report the formula used.

Line 137-139
there is a typo:
“BMI was classified as normal if BMI was [< ] 18.5 - 24.9 kg/m2, while overweight and obesity if BMIs were 25–29.9 kg/138 m2, and ≥ 30 kg/m2 respectively.”
the “<” before 18.5 is unintended, indeed Underweight <18.5

Line 205: please, again, specify the “some other diabetes-related factors”

Lines: 242-244
“First, this study was the first study conducted in Iraq, to assess the association of HL with glycaemic control among respondents with T2DM, using the S-TOFHLA questionnaire, and that was a challenge to authors.”

To be the first is a plus and not a defect, please explain precisely in which way it is a limitation of your study.

Line 245: “Lastly, the sample size was limited”
Did you inquire about the drop-out?

line 238-239  “dietary intake and physical activity level may predict HbA1c and the percentage will be increased to 35% of the total variation in glycaemic control, but these factors did not include in this study.”
I would ask the reason for not taking into account these two fundamental variables.

Author Response

Author's Reply to the Review Report (Reviewer 1):

The article contains a lot of observations but is not a full study and the discussion/conclusions are overemphasizing the correlation of low health literacy and not controlled diabetes quite ignoring other components or trying to explain a possible correlation.

Response 1: The correlations with other diabetes factors are discussed accordingly as follows:

Respondents within the self-employed category were associated with an increase in HbA1c compared with those in the employee category. This study has not confirmed previous studies on the association between HbA1c and self-employed status (Abdullah et al., 2019; Wan et al., 2016). In the Moroccan study, the employment term has no significant predictor of HbA1c among patients with T2DM (Chetoui et al., 2020). Nevertheless, a study conducted among 325 adults with T2DM attending in Jimma University Teaching Hospital in Ethiopia showed that farmers with diabetes have inadequate glycemic control compared with unemployed patients. Self-employment was associated with more extensive social networks, more job control, and higher stress (Belkic, Landsbergis, Schnall, & Baker, 2004).

As well as in term of FBG was associated with poor glycemic control, this finding coincides with other study conducted in Iran among 604 patients with T2DM, found a strong association between HbA1c and FBG and FBG is an accurate predictor for HbA1c (Ghazanfari, Haghdoost, Alizadeh, Atapour, & Zolala, 2010). This similarity in finding could be explained by the lack of control of FBG in daily life lead to poor glycemic control among patients with T2DM. 

Line 3: the title please, change “Other Diabetes-related Factors” with something more precise

Response 2: “Some Other Predicors” instead of  “Other Diabetes-related Factors”

Lines 83-84

“Respondents aged 20-64 years old, without hearing or vision impairment and severe illnesses like cancers were included in the study. They were excluded if they had type 1 diabetes, gestational diabetes, renal failure, or underwent active treatment for cancers.”
these two phrases should be rephrased in one clearer, simple and not redounded sentence.

Response 3:

Respondents aged 20-64 years, who had no hearing or vision impairment or severe illnesses such as cancer were included in the study. Whereas, patients with type 1 diabetes, gestational diabetes, or kidney failure were excluded.

Lines 91-93:

So you have chosen the twelfth subject counted from a random number for a total of eight individuals (samples) per day? please rephrase your sentences

Response 4:

According to the sample size, eight respondents were needed to be recruited daily during routine visits. Hence, we included the 12th patient from the list as a starting point generated by SPSS.

Line 93-94: “The sample size was determined using a specific formula (ref###) for a cross-sectional survey...” please cite and report the formula used.

Response 5:

The sample size was determined using a specific formula used in a cross-sectional study conducted by Daniel (Daniel, 2009), based on the prevalence of T2DM (19.7%) reported among respondents with T2DM in Basrah City (Mansour, Al-Maliky, Kasem, Jabar, & Mosbeh, 2014) as the following:

Line 137-139 there is a typo:

“BMI was classified as normal if BMI was [< ] 18.5 - 24.9 kg/m2, while overweight and obesity if BMIs were 25–29.9 kg/138 m2, and ≥ 30 kg/m2 respectively.”
the “<” before 18.5 is unintended, indeed Underweight <18.5

Response 6:

The “<” was removed.

Line 205: please, again, specify the “some other diabetes-related factors”

Response 7:

The “some other diabetes-related factors” changed to “ some other predictors”

Lines: 242-244

“First, this study was the first study conducted in Iraq, to assess the association of HL with glycaemic control among respondents with T2DM, using the S-TOFHLA questionnaire, and that  was a challenge to authors.”

To be the first is a plus and not a defect, please explain precisely in which way it is a limitation of your study.

Response 8:

Little experience with the S-TOFHLA scale among respondents with T2DM poses an obstacle to the authors.

Line 245:

“Lastly, the sample size was limited”
Did you inquire about the drop-out?

Response 9: Yes, the drop-out was 20%.

line 238-239  “dietary intake and physical activity level may predict HbA1c and the percentage will be increased to 35% of the total variation in glycaemic control, but these factors did not include in this study.” 

I would ask the reason for not taking into account these two fundamental variables.

Response 10:

These variables will be included in an independent study.

Reviewer 2 Report

Changing lifestyles and social stress lead to many problems in our lives. In particular, it plays a major role in causing diseases and therefore affecting humans. Especially, Diabetes is a very dangerous, multi-linked complications that plays a major role in the devastation of large numbers of people. In this manuscript, Saman Agad Hashim et al evaluates the health literacy connotation with glycemic control and other factors in type 2 Diabetes mellitus. The author has profiled 280 patients for their study. In short, the aim of the study is clear. The manuscript is well studied and considered to be very important, however the reviewer have few concerns as below will help to improve the quality of the manuscript.

Line 90, elaborate FDEMC in entire article

Line 94, include the specific formula for sample size determination

Line 284, Table A2, Authors has to check HbA1c for every three month interval to get accurate level because of RBC lifespan. How many times HbA1c checked? Many factors will affect HbA1C results, whether authors include the details about their medication in the questionnaire?

Did all patients aware about fasting blood glucose protocol? Is it included in the questionnaire?

Why authors specifically record lipid profile, no other profiles etc? Do author have any record other than Lipid?

Table A2 clearly indicates that there is no major changes in the biochemical parameters in HL from inadequate to adequate, this data limits the conclusion made.

Line 203, Discussion part, authors should discuss why Saudi Arabia study is inconsistent with another study?

In addition, include how can improve and raise awareness for HL to patients and others?

The authors have sufficient references to support their data and the text are readable.

Author Response

Author's Reply to the Review Report (Reviewer 2):

Changing lifestyles and social stress lead to many problems in our lives. In particular, it plays a major role in causing diseases and therefore affecting humans. Especially, Diabetes is a very dangerous, multi-linked complications that plays a major role in the devastation of large numbers of people. In this manuscript, Saman Agad Hashim et al evaluates the health literacy connotation with glycemic control and other factors in type 2 Diabetes mellitus. The author has profiled 280 patients for their study. In short, the aim of the study is clear. The manuscript is well studied and considered to be very important, however the reviewer have few concerns as below will help to improve the quality of the manuscript.

Line 90, elaborate FDEMC in entire article

Response 1: Faiha Specialized Diabetes, Endocrine, and Metabolism Centre (FDEMC) elaborated in the entire manuscript.

Line 94, include the specific formula for sample size determination

Response 2:

The sample size was determined using a specific formula used in a cross-sectional study conducted by Daniel (Daniel, 2009), based on the prevalence of T2DM (19.7%) reported among respondents with T2DM in Basrah City (Mansour, Al-Maliky, Kasem, Jabar, & Mosbeh, 2014) as the following:

Line 284, Table A2, Authors has to check HbA1c for every three month interval to get accurate level because of RBC lifespan. How many times HbA1c checked? Many factors will affect HbA1C results, whether authors include the details about their medication in the questionnaire?

Response 3:

Since the study was a cross-sectional study, the most recent HbA1c was taken from medical records.   The medication were included in the questionnaire.

Did all patients aware about fasting blood glucose protocol? Is it included in the questionnaire?

Response 4:

Yes, all respondents were aware about fasting blood glucose protocol and that was included in questionnaire.

Why authors specifically record lipid profile, no other profiles etc? Do author have any record other than Lipid?

Response 5:

As mentioned in several studies the lipid profile associated with poor glycemic control among patients with T2DM. renal function tests (urea and creatinine) were included in data collection but not in this study.

Table A2 clearly indicates that there is no major changes in the biochemical parameters in HL from inadequate to adequate, this data limits the conclusion made.

Response 6:

The data were reported as it is.

Line 203, Discussion part, authors should discuss why Saudi Arabia study is inconsistent with another study?

Response 7:

The disparity in findings is due to the developed health system in Saudi Arabia compared to Iraq.

In addition, include how can improve and raise awareness for HL to patients and others?

Response 8:

A simplified educational intervention is required to improve and raise the level of HL among patients and reduce complications or prevent disease among people.

The authors have sufficient references to support their data and the text are readable.

Reviewer 3 Report

This manuscript studies the association of health literacy with glycemic control in T2DM adults in a single-center, in Basrah, Iraq. The authors found that adequate healthy literacy is associated with the better glycemic control in patients. This point is significative and valuable, especially for disease like diabetes which requires the self-care or self-management of patients. However, I have several questions in term of this study.

  1. Could you specify the novelty of this study in title and abstract? It seems like the association between health literacy with glycemic control has been studied widely worldwide as the authors mentioned. And even in Iraq, there was similar studies.
  2. This study is using S-TOFHLA test. Could you explain more and compare the S-TOFHLA with other commonly used methods?
  3. What methods have been used by similar study in other locations/countries?
  4. Since this is a single-center study, could you compare your result with previous study also using S-TOFHLA test in different cities or centers?

Author Response

Author's Reply to the Review Report (Reviewer 3):

This manuscript studies the association of health literacy with glycemic control in T2DM adults in a single-center, in Basrah, Iraq. The authors found that adequate healthy literacy is associated with the better glycemic control in patients. This point is significative and valuable, especially for disease like diabetes which requires the self-care or self-management of patients. However, I have several questions in term of this study.

  1. Could you specify the novelty of this study in title and abstract? It seems like the association between health literacy with glycemic control has been studied widely worldwide as the authors mentioned. And even in Iraq, there was similar studies.

Response 1:

The title refers to some other predictors, and in the abstract, little studies used the Arabic version of S-TOFHLA with the same age group. There are some other variables mentioned in detail, like employment and treatment modalities. There is only one study conducted among pharmacies’ customers in Iraq, and the other used different instruments that self-developed by the author, which was not adequately validated and has not covered the whole spectrum of HL measures, as mentioned in the introduction.

  1. This study is using S-TOFHLA test. Could you explain more and compare the S-TOFHLA with other commonly used methods?

Response 2:

The most widely used are the Rapid Estimate of Adult Literacy in Medicine (REALM), Diabetes Numeracy Test (DNT) and the Test of Functional Health Literacy Assessment (TOFHLA). These scales are commonly used in research; the REALM tests the pronunciation of medical words, thus, has limitations as a measure of the full concept of HL. The DNT focuses on numeracy, demand long time and high literacy. The TOFHLA is considered the best; it measures both reading and numeracy skills. There are numerous measures of HL; the most popular of them is a Short version of the Test of Functional HL in Adults (S-TOFHLA) that was developed by Baker et al. (Baker, Williams, Parker, Gazmararian, & Nurss, 1999). S-TOFHLA has been commonly used to assess HL and takes about 12 minutes to complete (Baker et al., 1999). S-TOFHLA assesses both the reading and numeracy skills of subjects. In turkey, a cross-sectional study was conducted among patients with T2DM to assess HL using S-TOFHLA. The S-TOFHLA scale was associated significantly with age, education, and general diabetes knowledge. (Eyuboglu & Schulz, 2016). The other scales measure the HL among non diabetes respondents.

  1. What methods have been used by similar study in other locations/countries?

Response 3: in Japan, the 14-item health literacy scale was used, in Spain, HLS-EU-Q47 questionnaire was used, in Saudi Arabia  REALM-R test was used, in Iran, DNT15 was used and Newest Vital Sign tool was used in Jamaica.

  1. Since this is a single-center study, could you compare your result with previous study also using S-TOFHLA test in different cities or centers?

Response 4:

As we mentioned above, this study is the first study conducted in Iraq among diabetes patients. Nevertheless, we have mentioned previous studies that were conducted in other cities among non-diabetes patients in Iraq.

Round 2

Reviewer 1 Report

The manuscript is readable and the references complete; The author pointed out that diseases like diabetes require the self-care or self-management of patients.
The novelty of the manuscripts is the mention of “some other variables” that other worldwide studies about the association between health literacy with glycemic control have not.  

For this reason, I stress the needing to be more specific in this definition:

  • “The study aimed to assess the association of health literacy (HL) with glycemic control and some other predictors in[*]adults with type 2 diabetes mellitus (T2DM)”
    In my opinion, at least the first time in the abstract, the first in the main text, and finally in the conclusive remark, you should LIST what are these “some other predictors” Otherwise, your manuscript will sound vague and imprecise.
    By the way, line 19: “inadults” there is a space missing.
  • Sampling and Sample size: okay, the authors fixed this point, specifying the formula and collection modality of responders.
    thank you
  • I am wondering, why don’t you insert in the discussion a comparison among the S-TOFHLA with other commonly used methods (The most widely used are the Rapid Estimate of Adult Literacy in Medicine (REALM), Diabetes Numeracy Test (DNT), and the Test of Functional Health Literacy Assessment (TOFHLA)) as you already replied to the rev. #3?
  • Finally, I suggest to stress more in the conclusions remark, what you pointed out “diabetes requires the self-care or self-management of patients“

Author Response

Author's Reply to the Review Report (Reviewer 1, Round 2):

For the reason, I stress the needing to be more specific in this definition:

The study aimed to assess the association of health literacy (HL) with glycemic control and some other predictors in adults with type 2 diabetes mellitus (T2DM).

In my opinion at least the first time in the abstract, the first in the main text, and finally in the conclusive remark, you should LIST what these "some other predictors are"

Respond: Respond: Thank you very much for your comment. We realised the needs to have a specific definition for the ‘some other predictors’. Indeed, after a detailed thought we realised that may also not the appropriate term to use in the study. We would like to suggest using nutritional status assessments. This is because the nutritional status assessments include assessment of client history, anthropometry data, biochemical data, food-and nutrition related history and physical findings assessments. In this study, we collected data for client history, anthropometry data and biochemical data. We highlighted that inclusion of food and nutrition-related history and physical activity may enhance the prediction model to glycemic control.

We have corrected accordingly all term from some other predictors to nutritional status assessment as highlighted below and in the manuscript.

  • In the title line 2-3 as below:

we listed "Nutritional status assessments" which includes or means client history, glycemic control, anthropometric and biochemical data.

  • In the abstract line 36 as below:

we listed "Nutritional status assessments" which includes or means client history, glycemic control, anthropometric and biochemical data.

  • In the introduction line 86 as below:

we listed "Nutritional status assessments" which includes or means client history, glycemic control, anthropometric and biochemical data.  

  • In the methodology line 109-111 and 158 as below:

we listed "Nutritional status assessments" which includes or means client history, glycemic control, anthropometric and biochemical data.

  • In the results line 219 as below:

we listed "Nutritional status assessments" which includes or means client history, glycemic control, anthropometric and biochemical data.  

  • In the discussion line 244 as below:

we listed "Nutritional status assessments" which includes or means client history, glycemic control, anthropometric and biochemical data.  

  • In the conclusion line 315 we listed "Nutritional status assessments" which includes or means client history, glycemic control, anthropometric and biochemical data.

By the way, line 19: "inadults" there is a spacing missing.

Respond: the spacing missing added.

I wondering why don't you insert in the discussion a comparison among the S-TOFHLA with other commonly used methods the most widely used are the Rapid Estimate of Adult Literacy (REALM), the Diabetes Numeracy Test (DNT) and the Test of Functional Health Literacy in Adults (TOFHLA)) as you already replied to the rev.#3?

Respond: In this study, we focused on the specific type of health literacy measure (S-TOFHLA), so we compared our finding with the same standard to detect the exact differences between our study and the others, principally the S-TOFHLA test consists of two timed parts: (i) numeracy, which assesses a patient's ability to use numerical skills to comprehend directions; and (ii) reading comprehension. As well as, the S-TOFHLA was used at least 71.4% of the time the instrument was used. S-TOFHLA the first HL measure that assessed numeracy alongside literacy (Duell et al., 2015).

Duell, P.; David, W.; Andre, M.N.R.; Debi, B. Optimal health literacy measurement for the clinical setting: A systematic review. Patient Educ. Couns. 2015, 1-13, http://dx.doi.org/10.1016/j.pec.2015.04.003.

Finally, I suggest to stress more in the conclusions remark, what pointed out "diabetes requires the self-care or self-management of patients"

Respond: In line 247-249, the practice of self-diabetes care which leads to easily navigating the healthcare system and that directs to good glycemic control. In line 278-279 to practice self-diabetes care that leads to a decrease in the stress on health care professionals in diabetes centres. In line 286-287 self-care activities, which can contribute to consistencies in lifestyle and treatment outcomes, as evidenced by several studies.

Reviewer 3 Report

The authors answered my questions well. Please revise your manuscript and put some of the answers in the manuscript for a better understanding of readers. For example, in the Introduction, you can describe clearly the commonly used test such as Rapid Estimate of Adult Literacy in Medicine (REALM), Diabetes Numeracy Test (DNT) and the Test of Functional Health Literacy Assessment (TOFHLA). In addition, you can clearly describe why you choose the S-TOFHLA. You can also cite the reference you mentioned in the answers.

Author Response

Author's Reply to the Review Report (Reviewer 3, Round 2):

Please revise your manuscript and put some of the answers in the manuscript for a better understanding of readers. For example, in the introduction you can describe clearly the commonly used test such as the Rapid Estimate of Adult Literacy (REALM), the Diabetes Numeracy Test (DNT) and the Test of Functional Health Literacy in Adults (S-TOFHLA). In addition you can clearly describe why you choose the S-TOFHLA. You can also cite the references you mentioned in the answers.

Respond: Thank you for highlighting this valid comments. We have improved the paragraph by highlighting the critical component for each tool as written below.

HL is principally defined as 'the degree to which individuals can obtain, process, understand, and communicate about health-related information needed to make informed health decisions' (Nutbeam, 2008). There are various instruments used to assess HL in the diabetes population, such as the Rapid Estimate of Adult Literacy (REALM) (Davis et al., 1993), Diabetes Numeracy Test (DNT) (White, Osborn, Gebretsadik, Kripalani, & Rothman, 2011), and Test of Functional Health Literacy in Adults (S-TOFHLA) (Schillinger et al., 2002), of which S-TOFHLA was the most commonly used and considered a suitable tool to evaluate HL among adults with T2DM (Al Sayah, Williams, & Johnson, 2013). TOFHLA can be self-administered to assess reading comprehension by determining a patient's ability to read phrases and passages containing numbers using real objects in health care settings (Parker, Baker, Williams, & Nurss, 1995).

References

Al Sayah, F., Williams, B., & Johnson, J. A. (2013). Measuring health literacy in individuals with diabetes: a systematic review and evaluation of available measures. Health Educ Behav, 40(1), 42-55. doi:10.1177/1090198111436341

Davis, T. C., Long, S. W., Jackson, R. H., Mayeaux, E. J., George, R. B., Murphy, P. W., & Crouch, M. A. (1993). Rapid estimate of adult literacy in medicine: a shortened screening instrument. Family Medicine, 25(6), 391–395.

Nutbeam, D. (2008). The evolving concept of health literacy. Soc Sci Med, 67(12), 2072-2078. doi:10.1016/j.socscimed.2008.09.050

Parker, M., Baker, D., Williams, M., & Nurss, J. (1995). The Test of functional Health Literacy in Adults: A new instrument for measuring patient's literacy skills. . J General Internal Medicine, 10, 537-541.

Schillinger, D., Grumbach, K., Piette, J., Wang, F., Osmond, D., Daher, C., . . . Bindman, A. B. (2002). Association of health literacy with diabetes outcomes. Jama, 288(4), 475-482.

White, R. O., Osborn, C. Y., Gebretsadik, T., Kripalani, S., & Rothman, R. L. (2011). Development and validation of a Spanish diabetes-specific numeracy measure: DNT-15 Latino. Diabetes Technol Ther, 13(9), 893-898. doi:10.1089/dia.2011.0070
